# Embryonic origin and serial homology of gill arches and paired fins in the skate, *Leucoraja erinacea*

Victoria A Sleight[1,2†], J Andrew Gillis[1,2]*

[1]Department of Zoology, University of Cambridge, Cambridge, United Kingdom;
[2]Marine Biological Laboratory, Woods Hole, United Kingdom

**Abstract** Paired fins are a defining feature of the jawed vertebrate body plan, but their evolutionary origin remains unresolved. Gegenbaur proposed that paired fins evolved as gill arch serial homologues, but this hypothesis is now widely discounted, owing largely to the presumed distinct embryonic origins of these structures from mesoderm and neural crest, respectively. Here, we use cell lineage tracing to test the embryonic origin of the pharyngeal and paired fin skeleton in the skate (*Leucoraja erinacea*). We find that while the jaw and hyoid arch skeleton derive from neural crest, and the pectoral fin skeleton from mesoderm, the gill arches are of dual origin, receiving contributions from both germ layers. We propose that gill arches and paired fins are serially homologous as derivatives of a continuous, dual-origin mesenchyme with common skeletogenic competence, and that this serial homology accounts for their parallel anatomical organization and shared responses to axial patterning signals.

**\*For correspondence:**
jag93@cam.ac.uk

**Present address:** [†]School of Biological Sciences, University of Aberdeen, Zoology Building, Tillydrone, Aberdeen, United Kingdom

**Competing interests:** The authors declare that no competing interests exist.

## Introduction

It was classically proposed that the paired fins of jawed vertebrates evolved by transformation of a gill arch – a theory stemming largely from Gegenbaur's (*Gegenbaur, 1878*) interpretation of a shared anatomical ground plan between the gill arch and pectoral fin skeletons of cartilaginous fishes (sharks, skates and rays) (reviewed by *Coates, 1994*; *Coates, 2003*). In vertebrate embryos, the jaw, hyoid and gill arch skeleton (or, in amniotes, their derivatives, the jaw, auditory ossicles and laryngeal skeleton) arises from a series of transient, bilaterally paired pharyngeal arches that form on the sides of the embryonic head (*Gillis et al., 2012a*; *Graham et al., 2019*), while the paired fins or limbs of jawed vertebrates arise as buds that project from the embryonic trunk (*Tickle, 2015*). Cell lineage tracing studies in bony vertebrates (*Chai et al., 2000*; *Jiang et al., 2002*; *Couly et al., 1993*; *Kague et al., 2012*) have revealed that the pharyngeal arch skeleton derives largely from the neural crest – a vertebrate-specific, multipotent cell population that undergoes epithelial-to-mesenchymal transition from the dorsal neural tube, and that gives rise to a plethora of derivatives, including skeletal and connective tissue lineages (*Green et al., 2015*). The skeleton of paired appendages, on the other hand, derives from the lateral plate – a distinct mesodermal subpopulation that arose along the chordate stem (*Prummel et al., 2019*; *Prummel et al., 2020*). As shared embryonic origin has classically been regarded as a key criterion for serial homology (discussed by *Hall, 1995*), Gegenbaur's gill arch hypothesis of paired fin origin was widely discounted, and is now generally deemed fundamentally flawed (*Coates, 2003*).

Importantly, though, the distinct embryonic origins of the gill arch and paired fin skeletons may not hold true: mesodermal contributions to the posterior pharyngeal skeleton have been demonstrated in tetrapods, but are much less widely appreciated than those from neural crest. Cell lineage tracing using quail-chick chimaeras and viral labelling have revealed that the avian cricoid and arytenoid laryngeal cartilages derive from lateral mesoderm, and not neural crest (*Noden, 1986*;

**eLife digest** A common way to evolve new body parts is to copy existing ones and to remodel them. In insects for example, the antennae, mouth parts and legs all follow the same basic body plan, with modifications that adapt them for different uses. In the late 19th century, anatomist Karl Gegenbaur noticed a similar pattern in fish. He saw similarities between pairs of fins and pairs of gills, suggesting that one evolved from the other. But there is currently no fossil evidence documenting such a transformation.

Modern research has shown that the development of both gill and fin skeletons shares common genetic pathways. But the cells that form the two structures do not come from the same place. Gill skeletons develop from a part of the embryo called the neural crest, while fin skeletons come from a region called the mesoderm. One way to test Gegenbaur's idea is to look more closely at the cells that form gill and fin skeletons as fish embryos develop. Here, Sleight and Gillis examined the gills and fins of a cartilaginous fish called *Leucoraja erinacea*, also known as the little skate.

Sleight and Gillis labelled the cells from the neural crest and mesoderm of little skate embryos with a fluorescent dye and then tracked the cells over several weeks. While the fins did form from mesoderm cells, the gills did not develop as expected. The first gill contained only neural crest cells, but the rest were a mixture of both cell types. This suggests that fins and gills develop from a common pool of cells that consists of both neural crest and mesoderm cells, which have the potential to develop into either body part. This previously unrecognised embryonic continuity between gills and fins explains why these structures respond in the same way to the same genetic cues, regardless of what cell type they develop from. Based on this new evidence, Sleight and Gillis believe that Gegenbaur was right, and that fins and gills do indeed share an evolutionary history.

While firm evidence for the transformation of gills into fins remains elusive, this work suggests it is possible. A deeper understanding of the process could shed light on the development of other repeated structures in nature. Research shows that animals use a relatively small number of genetic cues to set out their body plans. This can make it hard to use genetics alone to study their evolutionary history. But, looking at how different cell types respond to those cues to build anatomical features, like fins and gills, could help to fill in the gaps.

*Noden, 1988*; *Evans and Noden, 2006*) – a finding that has since been corroborated by genetic lineage tracing experiments in mouse (*Tabler et al., 2017*; *Adachi et al., 2020*). Additionally, ablation (*Stone, 1926*) and lineage tracing experiments (*Davidian and Malashichev, 2013*; *Sefton et al., 2015*) have revealed a mesodermal origin of the posterior basibranchial cartilage in axolotl. Currently, however, there are no mesodermal fate maps of the pharyngeal skeleton of fishes, and so it remains to be determined whether mesodermal contributions to the posterior pharyngeal skeleton are an ancestral feature of jawed vertebrates, and whether mesoderm is competent to give rise to gill arch cartilages – that is, the ancestral skeletal derivatives of the posterior pharyngeal arches, and Gegenbaur's proposed evolutionary antecedent to paired fins.

We sought to map the contributions of neural crest and mesoderm to the pharyngeal and paired fin endoskeleton in a cartilaginous fish, the little skate (*Leucoraja erinacea*), as data from this lineage will allow us to infer ancestral germ layer contributions to the pharyngeal and paired fin skeletons, and to test the developmental potential of neural crest and mesodermal skeletal progenitors in a taxon that has retained an ancestral gill arch anatomical condition. We find that the gill arch skeleton of skate embryos receives contributions from both cranial neural crest and lateral mesoderm, revealing its dual embryonic origin. These findings point to an ancestral dual embryonic origin of the pharyngeal endoskeleton of jawed vertebrates, and to gill arches and paired appendages as serial derivatives of a dual-origin, neural crest- and mesodermally-derived mesenchyme with equivalent skeletogenic potential at the head-trunk interface.

## Results

### Neural crest and lateral mesoderm in the skate neurula

In the skate, neural tube closure begins at embryonic stage (S)16 and is complete by S18 (*Ballard et al., 1993*). in situ expression analysis of the gene encoding the conserved neural crest specifier Foxd3 reveals that by S18, pre-migratory cranial neural crest cells are specified within the dorsal neural tube but are not yet undergoing epithelial-to-mesenchymal transition (*Figure 1A*). At S18, we can also recognize molecularly distinct lateral mesodermal populations, including *tbx1-*

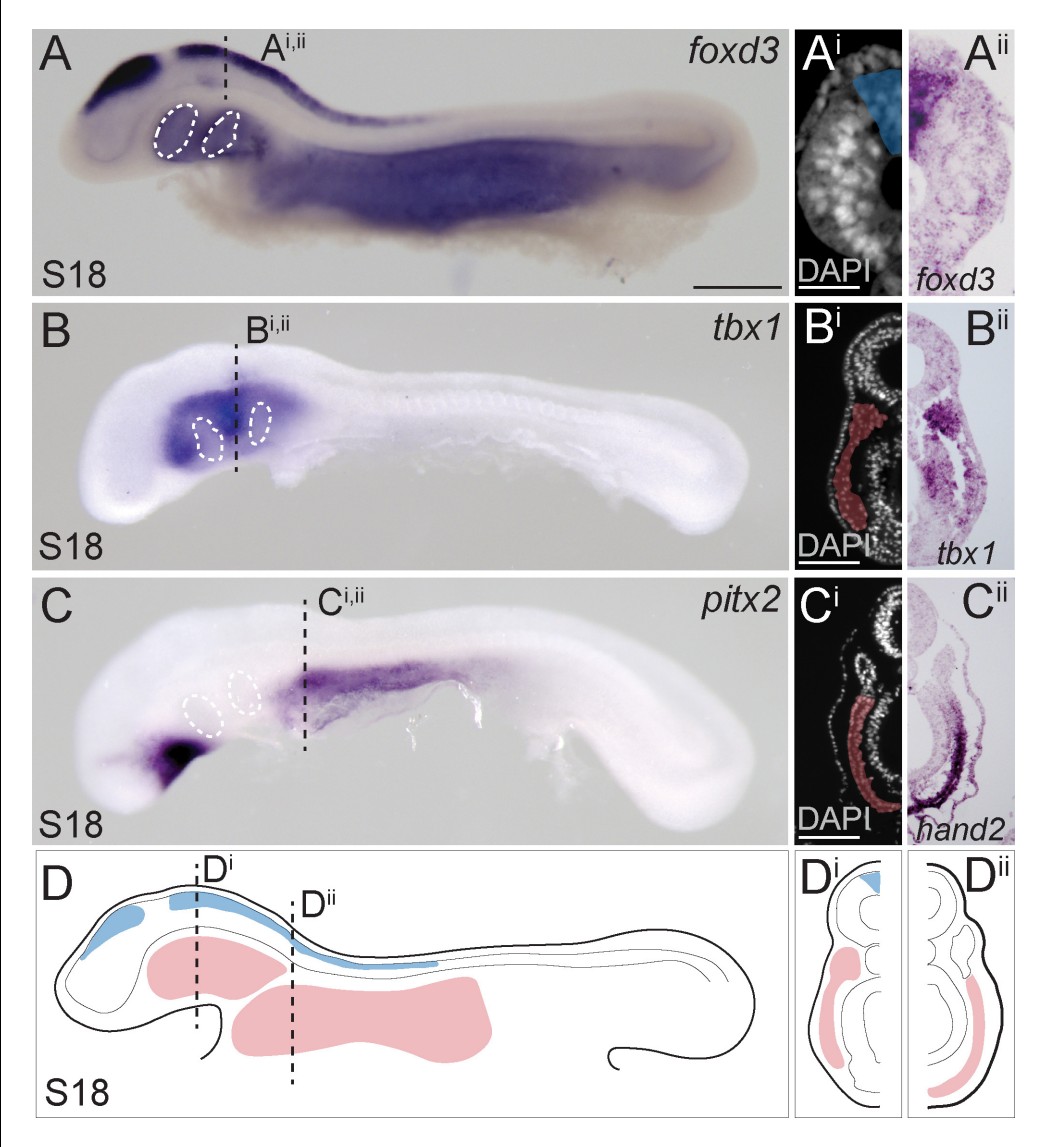

**Figure 1.** Neural crest and mesoderm after skate neurulation. (**A**) Wholemount mRNA in situ hybridization for *foxD3* reveals expression in (**Ai, Aii**) pre-migratory neural crest cells within the dorsal neural tube of the skate embryo at S18. (**B, Bi, Bii**) *tbx1*-expressing head mesoderm grades into (**C**) *pitx2*- and (**Ci, Cii**) *hand2*-expressing lateral plate mesoderm in the skate embryo at S18. (**D**) Schematic representation of neural crest, head mesoderm and lateral plate mesoderm tissues targeted for cell lineage tracing in this study. White dashed lines indicate the location of developing pharyngeal endodermal pouches. Scale bars: (**A, B**), C = 700 µm; Ai = 65 µm; (Bi), Ci = 120 µm.

The online version of this article includes the following figure supplement(s) for figure 1:

**Figure supplement 1.** Paraxial mesoderm in the little skate.

positive cranial paraxial mesoderm, or 'head mesoderm' (*Figure 1B*), which is morphologically continuous with *pitx2-* and *hand2*-positive lateral plate mesoderm (*Figure 1C*), and *myf5*-positive somitic and pre-somitic paraxial mesoderm (*Figure 1—figure supplement 1*). The clear spatial segregation and accessibility of these tissues in S18 skate embryos (*Figure 1D*) renders them amenable to fate mapping by labelling with lipophilic dyes – either by microinjecting the lumen of the neural tube (to label pre-migratory neural crest cells), or by microinjecting mesodermal mesenchyme underneath the head ectoderm – and so we used this approach to directly test the contributions of these tissues to the pharyngeal and paired fin endoskeleton.

## Neural crest contributes to the skate jaw, hyoid, and gill arch skeleton

To label skate cranial neural crest (NC) cells, we microinjected the lumen of the neural tube with CM-DiI at the hindbrain level. This resulted in very bright labelling at the point and time of injection (*Figure 2A*), though analysis of embryos collected shortly post-injection in section reveals that the cells of the neural tube were labelled around its entire circumference, broadly, along the length of the hindbrain region (*Figure 2A^i*). At five days post-injection, we observed abundant CM-DiI-labelled NC cells streaming into the pharyngeal arches (*Figure 2B*), and at S31/32 (~8–10 weeks post-injection), we tested for NC contributions to cartilages throughout the pharyngeal skeleton.

We have previously shown that the cartilaginous skeletal elements of embryonic skates can be readily identified, morphologically, in DAPI-stained vibratome or paraffin sections (*Figure 2—figure supplement 1*), and that labelling of early embryonic progenitors with lipophilic dyes is an effective way of mapping contributions to the cartilaginous endoskeleton (*Gillis et al., 2013*; *Gillis and Hall, 2016*; *Gillis et al., 2017*; *Criswell et al., 2017*; *Criswell and Gillis, 2020*). While the extent of CM-DiI-labelling of skeletal derivatives is always greatly reduced, relative to the labelling of progenitor cells at the time of injection (due to dilution of the CM-DiI label over several weeks of growth), positively-labelled cells are nevertheless unequivocally recognizable within the skeleton, due to the persistent brightness of the label. To add an additional level of stringency to our analysis, we only scored contributions to the skeleton consisting of clusters of two or more labelled cells, and contributions that were located in the centre of a skeletal element (to avoid inadvertently scoring CM-DiI-labelled connective tissue abutting the cartilage). As embryonic cartilage is a homogeneous tissue, consisting of a single cell type (the chondrocyte), we can therefore trace, with great certainty, the contributions of labelled progenitors to the differentiated cartilaginous endoskeleton.

Using the approach outlined above, we readily observed clusters of NC-derived chondrocytes, for example, in the cartilage of the palatoquadrate (*Figure 2C*) and the epibranchial and branchial ray cartilages of the first gill arch (*Figure 2D*). Overall, our analysis recovered NC contributions to major paired elements of the pharyngeal skeleton (i.e. jaw, hyoid and gill arch elements) and/or to the ventral midline cartilages across all labelled embryos (n = 20/20), but no contributions to the pectoral girdle (*Figure 2E*; *Supplementary file 1*). These findings are consistent with previous assessments of NC contribution to the pharyngeal and paired fin skeleton of zebrafish using genetic lineage tracing (*Kague et al., 2012*).

## Lateral mesoderm contributes to the skate gill arch and pectoral fin skeleton

We next sought to complement our NC fate map with a test for mesodermal contributions to the pharyngeal and paired fin skeleton in the skate. To do this, we used sub-ectodermal microinjection of lipophilic dyes (CM-DiI or SpDiOC$_{18}$) to label lateral mesoderm at three positions – within the *tbx1*-expressing head mesoderm (HM), at the boundary between HM and *pitx2/hand2*-expressing lateral plate mesoderm (LPM), or exclusively within LPM (*Figure 1D*) – either alone (*Figure 3A*), or in combination with neural crest labelling (*Figure 3B*). We once again left labelled embryos to develop for ~8–10 weeks post-injection, and then scored the embryos for contributions to the skeleton, as described above. Embryos labelled within the HM at S18 showed little contribution to the pharyngeal skeleton (labelled chondrocytes were recovered in gill arch cartilage of n = 1/10 labelled embryos; *Supplementary file 1*), while in the collective majority of embryos labelled at the HM-LPM boundary (n = 10/21) or within the LPM (n = 14/17), we observed substantial contributions to the pectoral girdle and fin skeleton (*Figure 3C*). The 'cardiopharyngeal field' is a mesodermal territory that encompasses both the cranial paraxial and anterior lateral plate mesoderm, and that gives rise

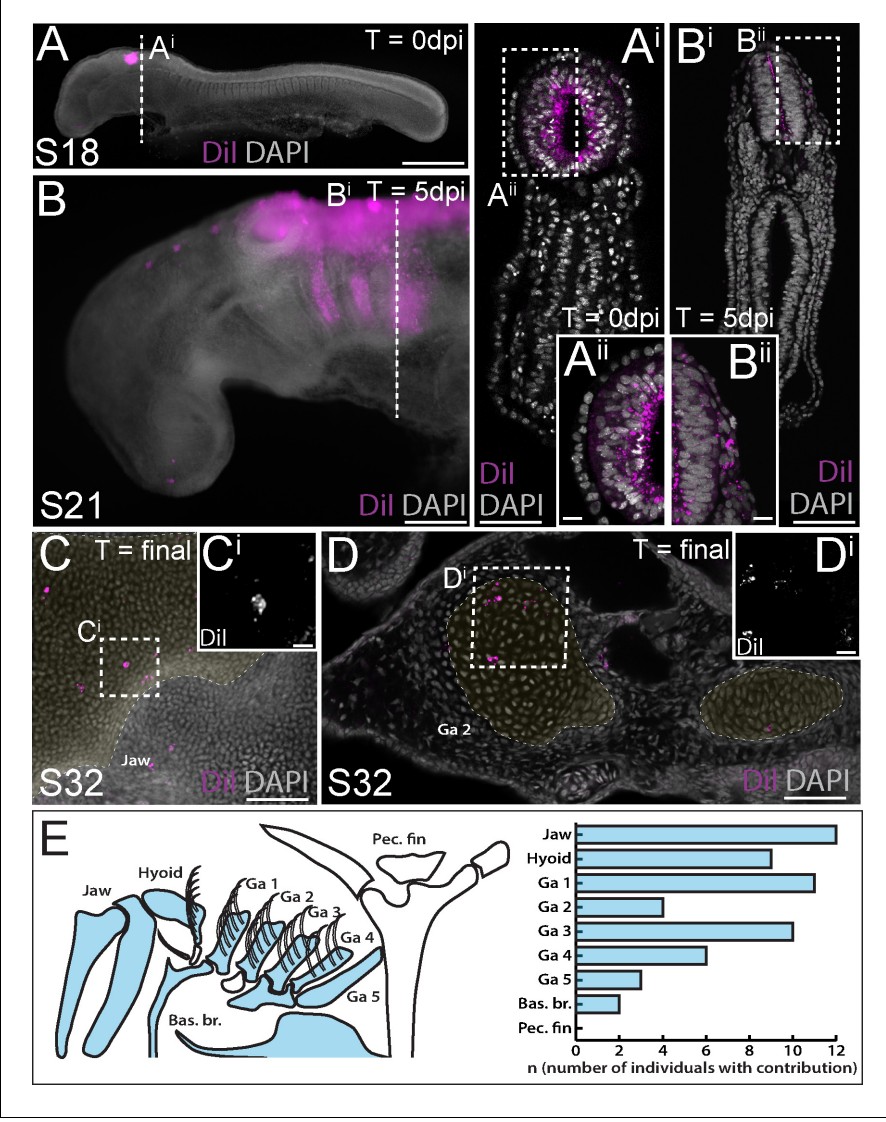

**Figure 2.** Neural crest contributes to the jaw, hyoid and gill arch skeleton in the skate. (A) Microinjection of CM-DiI into the lumen of the neural tube at S18 results in (Ai, Aii) labelling of cells throughout the hindbrain neural tube, including premigratory neural crest cells. (B) At 5 days post-injection (dpi), CM-DiI-labelled cranial neural crest cells can be seen streaming from the hindbrain neural tube into the pharyngeal arches (see also Bi, Bii). At S32, CM-DiI (i.e. neural crest-derived) chondrocytes are recovered within pharyngeal arch skeletal elements, including (C, Ci) the palatoquadrate of the jaw and (D, Di) the epibranchial of gill arch 2. (E) Schematic representation of pharyngeal and pectoral fin skeletal elements in the S32 skate embryo, with elements receiving contribution from neural crest coloured blue, and a plot showing the number of embryos observed with neural crest contributions to the pharyngeal arch skeleton. In (C and D), cartilaginous elements are false-coloured yellow. Scale bars: A = 700 µm; Ai = 250 µm; Aii = 50 µm; B = 340 µm; Bi = 250 µm; Bii = 50 µm; C = 165 µm; Ci = 15 µm; D = 70 µm; Di = 20 µm.

The online version of this article includes the following figure supplement(s) for figure 2:

**Figure supplement 1.** Identification of skate pharyngeal arch skeletal elements in section.

to pharyngeal arch (branchiomeric) musculature and the cardiovascular system (*Diogo et al., 2015*; *Prummel et al., 2020*). Accordingly, we observed extensive contributions from the skate HM, HM-LPM and LPM domains to the heart, blood vessels and pharyngeal arch musculature (*Figure 3—figure supplement 1*; *Supplementary file 1*).

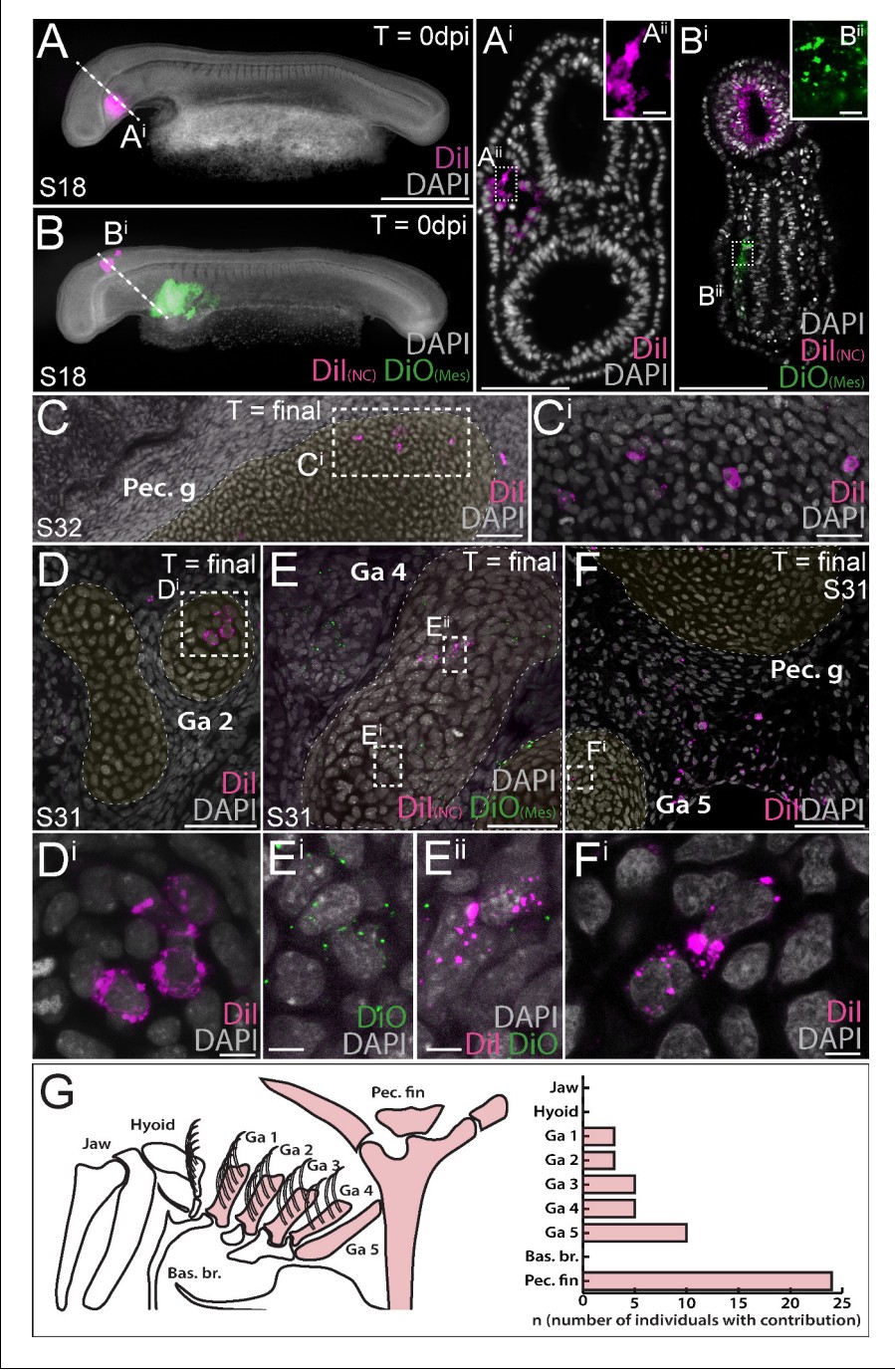

**Figure 3.** Mesoderm contributes to the gill arch and pectoral fin skeleton in the skate. (A, Aⁱ) Microinjection of CM-DiI into the head mesoderm (HM) of a skate embryo at S18. (B, Bⁱ) Simultaneous labelling of the hindbrain neural tube (including premigratory cranial neural crest cells) with CM-DiI and lateral plate mesoderm (LPM) with SpDiOC$_{18}$ in a S18 skate embryo. (C, Cⁱ) LPM gives rise to chondrocytes within the skeleton of the pectoral fin and girdle, while mesoderm at the HM-LPM boundary and LPM give rise to chondrocytes within the gill arch skeleton – e.g. (D, Dⁱ) in the branchial rays of gill arch 2. (E) After double labelling of the LPM with SpDiOC$_{18}$ and the neural tube with CM-DiI, as in (B) above, both (Eⁱ) SpDiOC$_{18}$- and (Eⁱⁱ) CM-DiI-labelled chondrocytes are recovered within the gill arch skeleton – for example, in the ceratobranchial of gill arch 4 – demonstrating the dual mesodermal and neural crest origin of these elements. (F, Fⁱ) Mesodermally-derived chondrocytes were also recovered in the ceratobranchial of gill arch 5, in close proximity to the label-retaining pectoral girdle and surrounding connective tissue. (G) Schematic summary of pharyngeal and paired fin skeletal elements in the S32 skate embryo, with

*Figure 3 continued on next page*

*Figure 3 continued*

elements receiving any mesodermal contributions (HM, HM-LPM or LPM) coloured red, and a plot showing the number of embryos observed with mesoderm contributions to the pharyngeal arch and pectoral fin skeleton. In (D), (E) and (F), cartilaginous elements are false-coloured yellow. Scale bars: A, B = 700 µm; A$^i$ = 125 µm; A$^{ii}$ = 15 µm; B$^i$ = 50 µm; B$^{ii}$ = 15 µm; C = 60 µm; C$^i$ = 20 µm; D = 50 µm; D$^i$ = 5 µm; E = 30 µm; E$^i$ = 5 µm; F = 60 µm; F$^i$ = 7 µm.

The online version of this article includes the following figure supplement(s) for figure 3:

**Figure supplement 1.** Cardiopharyngeal mesodermal derivatives in the skate.

**Figure supplement 2.** Dual embryonic neural crest and mesodermal origin of gill arch cartilages in the skate.

Remarkably, in many embryos labelled at the HM-LPM boundary (n = 11/21) or within LPM (n = 8/17), we also recovered label-retaining chondrocytes in the skeleton of gill arches 1–5. Mesodermally-derived chondrocytes were recovered within the epi- or ceratobranchial cartilages and branchial rays of gill arches 1–4 (e.g. *Figure 3D,E*), as well as in the ceratobranchial of gill arch 5, in close proximity to the label-retaining pectoral girdle and surrounding connective tissue (*Figure 3F* – also, see *Figure 3—figure supplement 2* for additional examples of mesoderm-derived label-retaining chondrocytes within the gill arch skeleton). Overall, our analysis recovered no mesodermal contributions to the mandibular or hyoid arch skeleton, but substantial mesodermal contributions to the paired cartilages of gill arches 1–5, as well as to the pectoral girdle and fin skeleton (*Figure 3G*; *Supplementary file 1*).

## Discussion

When considered alongside lineage tracing data from bony fishes, our findings allow us to infer an ancestral mesodermal contribution to the jawed vertebrate gill arch skeleton (*Figure 4A*), with the transition from neural crest-derived to mesodermally-derive skeletogenic mesenchyme occurring gradually, and spanning the region of the posterior (i.e. ancestrally gill-bearing) pharyngeal arches (*Figure 4B*). Taken together, our fate mapping experiments point to a neural crest origin of the mandibular and hyoid arch skeleton, a dual NC/mesodermal origin of the gill arch skeleton and an exclusively mesodermal origin of the pectoral fin skeleton in cartilaginous fishes (*Figure 4C*). In light of the dual embryonic origin of the mammalian thyroid cartilage and exclusively mesodermal origin of the cricoid and arytenoid cartilages (which are regarded as derivatives of the 4[th] and 6[th] pharyngeal arches), it is likely that boundaries of neural crest- and mesodermally-derived skeletogenic mesenchyme have shifted through vertebrate evolution.

Our findings also have important implications for understanding the evolutionary origin of paired appendages. With waning support for Gegenbaur's gill arch hypothesis, the lateral fin fold hypothesis of Balfour (*Balfour, 1881*), Thacher (*Thacher, 1877*) and Mivart (*Mivart, 1879*) emerged as the favoured scenario of paired fin origins. This hypothesis purports that paired fins originated from a continuous epithelial fold that flanked the trunk of the embryo, and that was subsequently segmented into distinct appendages at the pectoral and pelvic levels (reminiscent of the origin of the 1[st] and 2[nd] dorsal fins from a continuous median fin fold in sharks). While palaeontological and embryological evidence for the existence of a lateral fin fold (in phylogeny or ontogeny) remains scant, there is evidence of shared molecular patterning mechanisms between dorsal median fins and paired appendages (*Freitas et al., 2006*; *Dahn et al., 2007*; *Letelier et al., 2018*), and of the existence of broad zones of competence along the length of the trunk, from which ectopic fin/limbs or buds may be induced to form (*Cohn et al., 1995*; *Kawakami et al., 2001*; *Yonei-Tamura et al., 2008*). From these observations, a scenario has emerged in which an established appendage patterning developmental module was co-opted, bilaterally, from the dorsal midline to the flank, giving rise to paired pectoral and pelvic appendages.

We previously discovered shared, biphasic roles for Shh signalling in anteroposterior axis establishment and proliferative expansion of skeletal progenitors in the skate hyoid and gill arches and the tetrapod limb bud (*Gillis et al., 2009*; *Gillis and Hall, 2016*), and we now show that these shared patterning functions transcend the germ layer origin of Shh-responsive skeletogenic mesenchyme (i.e. neural crest alone in the hyoid arch, neural crest and lateral mesoderm in the gill arches and lateral mesoderm alone in the fin/limb bud) (*Figure 4B*). We propose that shared responses of hyoid,

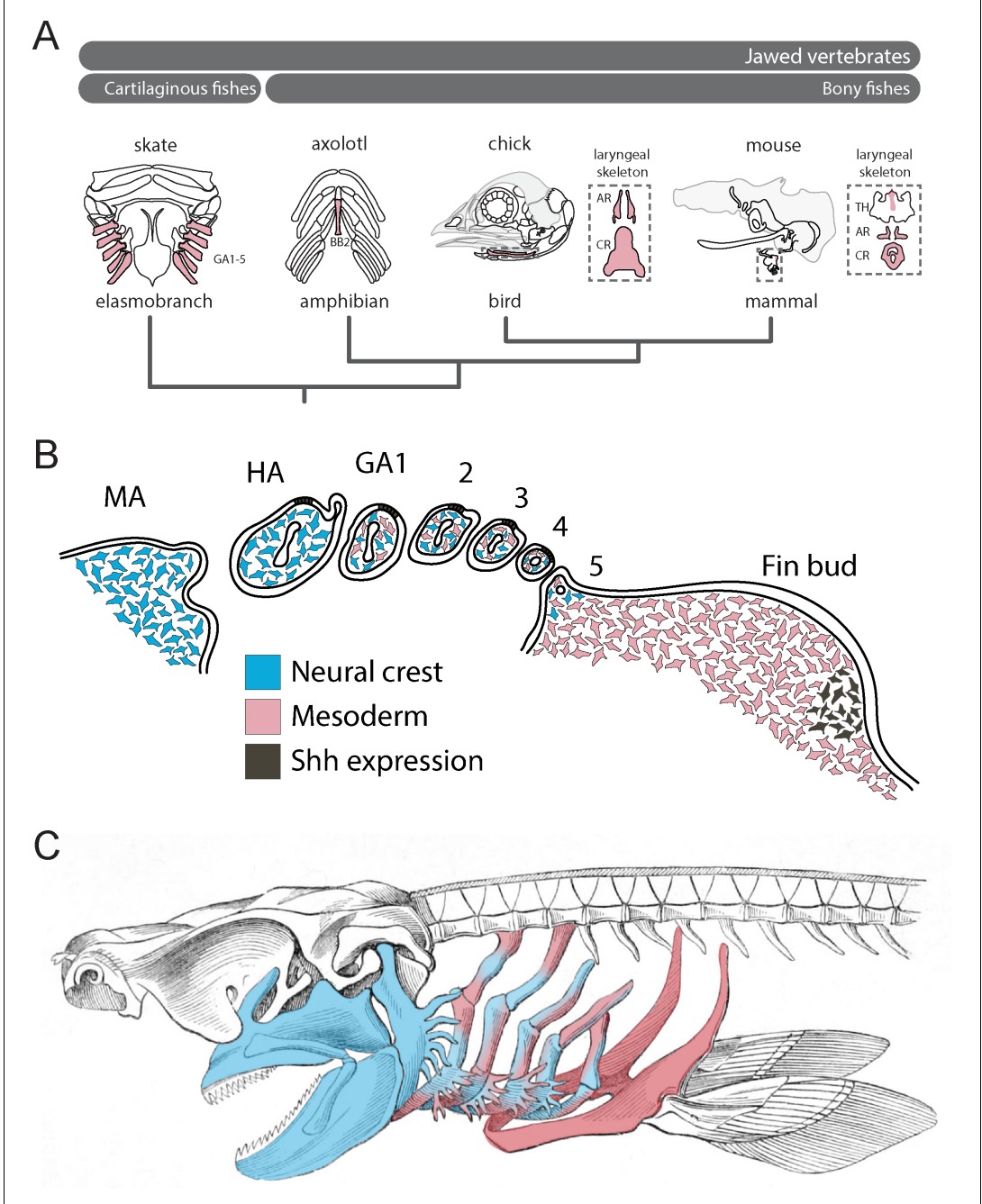

**Figure 4.** Mesodermal contributions to the pharyngeal endoskeleton in jawed vertebrates. (A) Mesodermal contributions (red) to the gill arch skeleton in skate, the basibranchial skeleton in axolotl and the laryngeal skeleton of chick and mouse points to an ancestral mesodermal contribution to the pharyngeal arch skeleton of jawed vertebrates. (B) Schematic representation of neural crest- (blue) and mesoderm-derived (red) skeletogenic mesenchyme in the skate pharyngeal arches and pectoral fin bud, in relation to epithelial and mesenchymal *Shh* expression, respectively. (C) We propose that the mandibular and hyoid arch skeleton are neural crest-derived and the pectoral fin skeleton mesodermal derived, while the gill arch skeletal elements are of dual neural crest and mesodermal origin.

gill arch and limb skeletal elements to perturbations in Shh signalling – despite differences in the source of Shh in these organs (i.e. the gill arch epithelial ridge and limb bud zone of polarizing activity – *Riddle et al., 1993*; *Gillis and Hall, 2016*; *Figure 4B*) – reflect a common underlying competence of gill arch and fin/limb skeletogenic mesenchyme to respond to these patterning signals, and serial homology of the skeletal derivatives of this mesenchyme. The zones of competence that

underlie the origin of pectoral and pelvic appendages within the trunk could, accordingly, be extended rostrally to include zones of neural crest and mixed neural crest/lateral mesodermal contribution to the pharyngeal endoskeleton, and this, in turn, could account for the serial derivation of gill arches and paired appendages along the gnathostome stem. Indeed, reports of a fossil jawless vertebrate with gill arches extending down the length of the trunk (*Janvier et al., 2006*) further support the shared competence of pharyngeal and lateral trunk mesenchyme to give rise to both gill arch and fin/limb skeletal elements.

It has been proposed that the neural crest acquired its skeletogenic potential by co-opting a chondrogenic gene regulatory network that arose, ancestrally, within mesoderm (*Meulemans and Bronner-Fraser, 2007*; *Cattell et al., 2011*) – a view that is further supported by the discovery of conserved molecular features of the developing neural crest and mesoderm-derived cartilages of vertebrates and the (presumably mesoderm-derived) cellular cartilages of some invertebrates (*Cole and Hall, 2004a*; *Cole and Hall, 2004b*; *Jandzik et al., 2015*; *Tarazona et al., 2016*). It is therefore to be expected that neural crest and mesodermal mesenchyme share fundamental molecular mechanisms of skeletogenesis. However, there is nevertheless a heterogeneity across mesenchymal subpopulations in their competence to respond to particular patterning cues. For example, in birds, specific regions of foregut endoderm are both necessary and sufficient for the specification of mandibular arch skeletal elements, but can only induce these elements to form from the neural crest mesenchyme that populates the mandibular arch (and not from the mesenchyme of the more caudal pharyngeal arches – *Couly et al., 2002*). Conversely, quail-chick heterotopic transplantation experiments have shown that midbrain-derived neural crest mesenchyme is competent to give rise to the pleurosphenoid of the lateral braincase wall, even though this element typically derives exclusively from paraxial mesoderm (*Schneider, 1999*). Examples such as these point to more cryptic domains of skeletogenic mesenchyme, with distinct competencies, that do not necessarily align with germ layer boundaries. While the molecular basis of this mesenchymal regionalization may not be known, such regions of shared competence may be operationally defined using cell lineage tracing or transplantation experiments, and may be further tested for shared transcriptional features (i.e. indicative of shared downstream effectors of common inductive cues, and the deployment of shared gene regulatory networks). We also propose that, on an evolutionary time scale, these regions of competence may be predisposed to the iterative deployment of developmental mechanisms, resulting in serial homology.

Importantly, a competence-based hypothesis of gill arch-fin serial homology decouples the origin and evolutionary histories of gill arches/paired appendages as anatomical structures and the molecular mechanism that direct their patterning – i.e. it accounts for the former, but leaves the latter open to further discourse around the deep homology of appendage patterning mechanisms within vertebrates or, more broadly, metazoans (*Shubin et al., 2009*). It is widely appreciated that, in animals, a relatively small number of developmental signalling pathways are used repeatedly, and in different combinations/contexts, to instruct the development of a great many embryonic tissues and organs. This, in turn, precludes the straightforward inference of homology of anatomical structures based on shared molecular patterning mechanisms (*Dickinson, 1995*). We argue that recognition of anatomical similarity due to common *response* to instructive cues within generative tissues, rather than focusing on the cues themselves, can allow us to bridge the gap between patterning mechanisms and morphology, and may provide a basis for inferring homology of morphology, even when considering structures that develop under the influence of upstream patterning mechanisms with complex and/or distinct evolutionary histories.

Homology is a hierarchical concept, and two complex features (e.g. organs) – which arise within the context of an embryonic tissue, by deployment of a gene regulatory network operating downstream of an inductive or patterning cue – may be homologous at one biological level of organization, while simultaneously non-homologous at another (*Hall, 2003*; *Wagner, 2014*). While reconstructing the evolutionary history (homology) of individual genes or gene regulatory network nodes is becoming increasingly straightforward, meaningfully testing the homology of putatively distantly-related structures at the anatomical level – whether historical homologues across taxa, or serial homologues within a taxon – has, in many cases, lingered as problematic. Developmental competence, or the cell-autonomous property that imparts on tissues an ability to respond to external stimuli (e.g. organizers and signalling centres) (*Waddington, 1947*), may represent a tangible means of linking upstream molecular developmental mechanisms with ultimate anatomical readouts

(*Spemann, 1915*). In light of the demonstrated lability of germ layer fates within the vertebrate skeleton (*Teng et al., 2019*), we suggest that, in the case of anatomy, competence – which is inherently testable, either by natural (i.e. evolutionary) or laboratory experimentation – may supersede germ layer origin as a primary criterion of homology.

## Materials and methods

### Embryo collection

*L. erinacea* eggs were obtained at the Marine Biological Laboratory (Woods Hole, MA, USA) and maintained in a flow-through seawater system at ~15℃ to the desired developmental stage. Embryos for mRNA in situ hybridization were fixed in 4% paraformaldehyde in phosphate-buffered saline (PBS) overnight at 4℃, rinsed three times in PBS, dehydrated into 100% methanol and stored at −20℃. Embryos injected with CM-DiI and SpDiOC$_{18}$ were fixed in 4% paraformaldehyde in PBS overnight at 4℃, rinsed three times in PBS, and stored in PBS + 0.02% sodium azide at 4℃.

### mRNA in situ hybridization

*L. erinacea* embryos were embedded in paraffin wax and sectioned at 8 μm thickness for mRNA in situ hybridization as previously described (*O'Neill et al., 2007*). Wholemount and paraffin chromogenic mRNA in situ hybridization experiments for *FoxD3* (GenBank accession number MN478366), *Tbx1* (GenBank accession number MT150581), *Pitx2* (GenBank accession number MT150579), *Hand2* (GenBank accession number MT150580) and *Myf5* (GenBank accession number MT150582) were performed as previously described (*O'Neill et al., 2007*) with modifications according to *Gillis et al., 2012b*.

### Fate mapping and imaging

Preparation and microinjection of CM-DiI and SpDiOC$_{18}$ was carried out as previously described (*Gillis et al., 2017*; *Criswell and Gillis, 2020*). After labelling, sealed eggs were returned to a flow-through seawater system at ~15℃ to the desired developmental stage, and then euthanized using an overdose of tricaine (1 g/L in seawater) prior to fixation. Labelled embryos to be analysed by vibratome sectioning were rinsed 3 × 5 min in PBS, embedded in 15% (w/v) gelatin in PBS and post-fixed in 4% paraformaldehyde in PBS for 4 nights at 4℃ before sectioning at 100 μm on a Leica VT1000S vibratome. Sections were then DAPI-stained (1 μg/mL), coverslipped with Fluoromount-G (Southern Biotech) and imaged on an Olympus FV3000 confocal microscope. Labelled embryos to be analysed by paraffin histology were embedded and sectioned as previously described (*O'Neill et al., 2007*).

## Acknowledgements

The authors thank Dr. Richard Schneider, Prof. David Sherwood, and the MBL Embryology Course for provision of lab space, Louise Bertrand and Leica Microsystems for microscopy support, the staff of the Marine Resources Center at the MBL for assistance with animal husbandry and Dr. Kate Criswell, Jenaid Rees, Christine Hirschberger and Dr. Kate Rawlinson for helpful discussion. This project benefited from technical advice from Dr. Matt Wayland and use of the Imaging Facility, Department of Zoology, supported by a Sir Isaac Newton Trust Research Grant (18.07ii(c)). This research was supported by a Royal Society University Research Fellowship (UF130182) and grants from the Leverhulme Trust (RPG-2016–373) and the University of Cambridge Sir Isaac Newton Trust (14.23z) to JAG, and by a Junior Research Fellowship from Wolfson College, Cambridge and Whitman Early Career Fellowship from the Marine Biology Laboratory to VAS.

## Additional information

### Funding

| Funder | Grant reference number | Author |
|---|---|---|
| Royal Society | UF130182 | Andrew Gillis |

| Leverhulme Trust | RPG-2016-373 | Andrew Gillis |
|---|---|---|
| Isaac Newton Trust | 14.23z | Andrew Gillis |
| Wolfson College, University of Cambridge | Junior Research Fellowship | Victoria A Sleight |
| Marine Biological Laboratory | Whitman Early Career Fellowship | Victoria A Sleight |

The funders had no role in study design, data collection and interpretation, or the decision to submit the work for publication.

### Author contributions

Victoria A Sleight, Data curation, Investigation, Methodology, Writing - review and editing, VAS contributed to experimental design, performed, analyzed and imaged all experiments, and prepared all figures. VAS interpreted the data, and reviewed and edited the manuscript; J Andrew Gillis, Conceptualization, Supervision, Investigation, Methodology, Writing - original draft, Project administration, Writing - review and editing, JAG conceived and oversaw the study, interpreted the data and wrote the manuscript

### Author ORCIDs

J Andrew Gillis (iD) https://orcid.org/0000-0003-2062-3777

### Ethics

Animal experimentation: All of the animals used in this study were handled according to approved institutional animal care and use committee (IACUC) protocols (#17-31, 18-32 and 19-34) of the Marine Biological Laboratory in Woods Hole.

### Decision letter and Author response

Decision letter https://doi.org/10.7554/eLife.60635.sa1
Author response https://doi.org/10.7554/eLife.60635.sa2

## Additional files

### Supplementary files

• Supplementary file 1. Fate mapping data for neural crest and mesodermal lineage tracing experiments. Raw scoring data showing contributions from the neural crest and mesoderm to the skate pharyngeal skeleton, and schematic of the skate pharyngeal skeleton. Skeletal elements abbreviated as follows: *BC*, basibranchial copula; *BH*, basihyal; *CB1-5*, ceratobranchials 1–5; *C-ph*; ceratopseudohyal; *EB1-5*, epibranchials 1–5; *E-ph*, epipseudohyal; *HB2*; hypobranchial 2; *HB3/4*, hypobranchial 3/4; *HM*, hyomandibula; *H-ph*, hypopseudohyal; *HT*, heart; *MK*, Meckel's cartilage; *PB1-4*, pharyngobranchials 1–4; *PG*, pectoral girdle; *PQ*, palatoquadrate; *SC*, spiracular cartilage.

• Transparent reporting form

### Data availability

All data generated or analysed during this study are included in the manuscript and supporting files.

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
