## [Decision Letter]

**Acceptance summary:**

Developmental data sets from cartilaginous fish are critical for understanding the early evolution of the vertebrate body plan. This paper provides clear lineage mapping evidence that in Little Skate, the gill arches have a dual embryonic origin, receiving contributions from both neural crest and lateral plate mesoderm. These observations provide insight into how distinct embryonic cell populations segregate at the head-trunk interface in a representative cartilaginous fish and fuel new hypotheses for the origin of the vertebrate appendicular system.

**Decision letter after peer review:**

Thank you for submitting your article "Embryonic origin and serial homology of gill arches and paired fins in the skate (*Leucoraja erinacea*)" for consideration by *eLife*. Your article has been reviewed by three peer reviewers, and the evaluation has been overseen by a Reviewing Editor and Didier Stainier as the Senior Editor. The following individuals involved in review of your submission have agreed to reveal their identity: Elizabeth M Sefton (Reviewer #2); Robert Cerny (Reviewer #3).

The reviewers have discussed the reviews with one another and the Reviewing Editor has drafted this decision to help you prepare a revised submission.

Summary:

In this manuscript, the authors characterize neural crest and mesoderm contributions to the pharyngeal region and pectoral fins of the Little Skate as a representative Chondrichthyan. They use these new findings, together with their previously published data sets on arch patterning, to argue for the serial homology of gill arch and paired fin skeletons.

The authors' experimental strategy involved first using molecular markers to verify the position of pre-migratory neural crest, head mesoderm, and lateral plate mesoderm populations in early stage skate embryos, followed by a series of targeted injections with lipophilic vital dyes (DiI or SpDiOC18) to test if these populations contribute to the pharyngeal skeleton and pectoral fin. Injected embryos were raised eight to ten weeks, sectioned, and the distribution of labeled cells examined. To minimize the risk of false positives, labeled cells were scored as contributing to a skeletal element only if the cells were centrally located within the element and present in a cluster of at least two.

The results demonstrate neural crest contributions to the jaw, hyoid, and gill arches, consistent with NC lineage analyses in other vertebrates.

Injections of the head mesoderm, lateral mesoderm at the head-trunk interface, and trunk lateral plate mesoderm were also performed. The central finding of these mapping experiments was that lateral plate mesoderm contributed to both the gill arches, and the pectoral girdle and fin skeleton. These data, together with the authors' neural crest maps, provide evidence for a dual embryonic origin of the gill arches in a chondrichthyan.

Although dilution by cell division limits the resolution of vital dye fate maps, the consensus among reviewers is that the figures and results are clear and convincing. Further, the use of the Little Skate to address how embryonic cell populations segregate during skeletal formation at the head trunk interface fills an important phylogenetic gap in our knowledge of vertebrate development, one that has broader implications for understanding the evolution of the vertebrate body plan. The findings are therefore of general interest to the field.

Essential revisions:

The Discussion is well written, but revision is necessary to address several points raised by reviewers.

1) Gegenbauer's gill arch hypothesis provides interesting historical context for this study. However, as the authors point out, modern developmental genetics has shown that germ layer origin as a criterion for assigning homology can be suspect since cell identity and functionality are due to gene regulatory network activity. Cells with a mesenchymal, stem-like phenotype (like neural crest and LPM) are especially prone to activate diverse GRNs. While the authors argue for developmental "competence" over germ layer origin as a criterion for homology, they should elaborate on how they define "competence" and "common response to instructive cues" (Discussion), and how this relates to GRN function. "Competence" is just one component of a cell's phenotype, and gives some insight into the cell surface receptors and transcriptional regulators active in that cell type (specifically those genes needed to respond to ligands secreted by an organizer tissue). But this is not necessarily indicative of the deployment of a homologous GRN. Ultimately, the authors present evidence that the GRN of LPM cells likely shares features with the GRN of cranial neural crest cells in little skate, because they can behave like NC when they occupy the pharyngeal arches. This is an interesting finding that complements and motivates future transcriptomic/functional genomic comparisons of these populations in skate and other vertebrates to test hypotheses of homology.

2) If possible, the authors should discuss any consistent trends in the distribution of neural crest versus mesodermal labeling of gill arch elements. For example, does labeled mesoderm trend towards the dorsal or lateral portions of skeletal elements? Do the authors think that neural crest and mesoderm mix evenly? Or given inherent limits in the resolution of vital dye fate maps by dilution from cell division, is it too difficult to discern with these methods? If discernible, any such trend should be compared with published data from bony fishes.

3) Cranial and lateral mesoderm are often termed the cardiopharyngeal field, as CM/LPM makes contributions to both the heart and cranial muscle. Given the increasing interest in the cardiopharyngeal field, do the authors see heart labeling in any of their specimens (presuming the heart has been sectioned along with the gill arches and pectoral fins)? If so, this would be worthwhile to include as a supplementary figure and in their supplementary file. This is not required if the authors do not have access to this data. If the heart is not labeled in cranial mesoderm injections, this would be an interesting correlation with the absence of midline skeletal element labeling.

---

## [Author Response]

Essential revisions:The Discussion is well written, but revision is necessary to address several points raised by reviewers.1) Gegenbauer's gill arch hypothesis provides interesting historical context for this study. However, as the authors point out, modern developmental genetics has shown that germ layer origin as a criterion for assigning homology can be suspect since cell identity and functionality are due to gene regulatory network activity. Cells with a mesenchymal, stem-like phenotype (like neural crest and LPM) are especially prone to activate diverse GRNs. While the authors argue for developmental "competence" over germ layer origin as a criterion for homology, they should elaborate on how they define "competence" and "common response to instructive cues" (Discussion), and how this relates to GRN function. "Competence" is just one component of a cell's phenotype, and gives some insight into the cell surface receptors and transcriptional regulators active in that cell type (specifically those genes needed to respond to ligands secreted by an organizer tissue). But this is not necessarily indicative of the deployment of a homologous GRN. Ultimately, the authors present evidence that the GRN of LPM cells likely shares features with the GRN of cranial neural crest cells in little skate, because they can behave like NC when they occupy the pharyngeal arches. This is an interesting finding that complements and motivates future transcriptomic/functional genomic comparisons of these populations in skate and other vertebrates to test hypotheses of homology.

We have included a new paragraph in our Discussion section which expands on our definition of “competence”, and which discusses our findings in the context of gene regulatory networks. Admittedly, we don’t yet know the nature of the GRN that is operating downstream of Shh signalling to pattern both the gill arch and paired fin/limb endoskeleton – indeed, this will be an important follow-up study to further test our hypothesis of serial homology (and aspects of this work are currently underway in my lab). Ultimately, we argue that domains of mesenchymal competence seem to transcend germ layer boundaries, and that such domains can currently be operationally defined and tested using embryological approaches (e.g. lineage tracing and transplantation). We provide additional published examples to support this view, and we also discuss how future work could take advantage of comparative transcriptomic approaches to discover shared downstream effectors of the signalling interactions that direct the fate of skeletogenic mesenchyme. We think that this new text addresses the excellent points that have been raised in the comment above.

2) If possible, the authors should discuss any consistent trends in the distribution of neural crest versus mesodermal labeling of gill arch elements. For example, does labeled mesoderm trend towards the dorsal or lateral portions of skeletal elements? Do the authors think that neural crest and mesoderm mix evenly? Or given inherent limits in the resolution of vital dye fate maps by dilution from cell division, is it too difficult to discern with these methods? If discernible, any such trend should be compared with published data from bony fishes.

Unfortunately, we aren’t able to speculate about the distribution of neural crest vs. mesodermal contributions to the gill arch skeleton. As the reviewer has pointed out, our lineage tracing methods are not quite of sufficiently high resolution to draw meaningful conclusions about this from our present study.

3) Cranial and lateral mesoderm are often termed the cardiopharyngeal field, as CM/LPM makes contributions to both the heart and cranial muscle. Given the increasing interest in the cardiopharyngeal field, do the authors see heart labeling in any of their specimens (presuming the heart has been sectioned along with the gill arches and pectoral fins)? If so, this would be worthwhile to include as a supplementary figure and in their supplementary file. This is not required if the authors do not have access to this data. If the heart is not labeled in cranial mesoderm injections, this would be an interesting correlation with the absence of midline skeletal element labeling.

This is a great point. We’d not previously discussed the concept of the “cardiopharyngeal field” in our manuscript, but we have now included a mention of this in our Results section. We did, indeed, observe extensive heart labeling in many of the embryos with HM, HM-LPM or LPM dye injections, and these data were recorded in our score sheets (though not reported in our original submission or included in our supplementary file). We have now included a statement about this in our revised manuscript (subsection “Lateral mesoderm contributes to the skate gill arch and pectoral fin skeleton”), and we have included mesodermal contributions to the heart in our Data table (Supplementary file 1). Additionally, we have removed the figure supplement showing CM-DiI-labeled pharyngeal musculature (originally Supplementary Figure 2) and have replaced this with a new supplementary figure (Figure 3—figure supplement 1) showing various cardiopharyngeal mesodermal derivatives (including heart, vascular endothelium and pharyngeal arch musculature).